# Flow Boiling Heat Transfer; Experimental Study of Hydrocarbon Based Nanorefrigerant in a Vertical Tube

**DOI:** 10.3390/nano13152230

**Published:** 2023-08-01

**Authors:** Marta Hernaiz, Iker Elexpe, Estibaliz Aranzabe, Andrés T. Aguayo

**Affiliations:** 1TEKNIKER Basque Research and Technology Alliance (BRTA), C/Iñaki Goenaga, 5, 20600 Eibar, Spain; iker.elexpe@tekniker.es (I.E.); estibaliz.aranzabe@tekniker.es (E.A.); 2Department of Chemical Engineering, University of the Basque Country (UPV/EHU), Apdo. 644, 48080 Bilbao, Spain; andrestomas.aguayo@ehu.eus

**Keywords:** two-phase flow boiling, vertical evaporator tube, nanoparticles, heat transfer coefficient, n-pentane, heat transfer mechanism

## Abstract

Flow boiling is a complex process but very efficient for thermal management in different sectors; enhancing flow boiling heat transfer properties is a research field of great interest. This study proposes the use of various nanomaterials, carbon-based materials, and metal oxides; in n-pentane as a hydrocarbon-based refrigerant to enhance the flow boiling heat transfer coefficient. This thermal property has been experimentally evaluated using a vertical evaporation device of glass with an internal diameter of 20 mm. The results have shown that proposed nanomaterials dispersion in n-pentane has a limited effect on the thermophysical properties and is conditioned by their dispersibility but promotes a significant increment of pentane heat transfer coefficient (h), increasing the overall heat transfer coefficient (U) of the evaporator. The enhanced heat transfer performance is attributed to the behavior of nanoparticles under working conditions and their interaction with the working surface, promoting a higher generation of nucleation sites. The observed behavior suggests a heat transfer mechanism transition from forced convection to nucleate heat transfer, supported by visual observations.

## 1. Introduction

The boiling process is a highly efficient heat transfer mechanism involving the phase change from liquid to vapor state. Flow boiling, which refers to the boiling of a moving stream within a channel, plays a crucial role in the thermal management of various engineered systems. It finds significant applications in both the industrial and domestic sectors due to its exceptional cooling efficiency. Flow boiling effectively addresses the challenge of dissipating large heat flow in both small-scale (mini-channels and micro-channels) [1,2,3,4,5] and large-scale (conventional/macro-channels) cooling applications [6,7,8,9]. Its industrial applications cover electronic and microelectronic devices, data centers, boiler tubes, and evaporators, as well as domestic applications, such as air cooling, refrigeration, and household appliances.

The two-phase heat transfer process (liquid–vapor phase change) is characterized by Collier and Thom’s flow boiling curve (Figure 1). This curve shows for heated vertical tubes the corresponding flow patterns and heat transfer mechanisms. The boiling heat transfer (h) is the relationship between the heat transfer surface heat flux, q″, and the degree of superheating of the heater surface, T_w_ − T_sat_, where T_w_ is the surface temperature, and T_sat_ is the saturation temperature corresponding to the flow system pressure.

As the excess temperature value increases, the curve passes through different regimes. At the inlet, the liquid enters subcooled (below saturation temperature); in this region, the flow is single-phase. As the liquid heats up, the wall temperature increases, and when it exceeds the saturation temperature, subcooled nucleate boiling begins. Further increase in liquid temperature causes the liquid bulk to reach its saturation temperature, and the convective boiling process passes through bubbly flow into slug flow. The increase in void fraction causes the flow structure to become unstable. The boiling process passes through the slug, and churns flow to the annular flow regime with its characteristic annular liquid film [10].

Boiling heat transfer, intensification has been proposed through different approaches, such as surface modifications (roughness [11] and porosity [12]) or geometrical modifications of the flow channel, such as the channel aspect ratio [13], as well as proposing refrigerants with enhanced thermophysical properties [14].

One of the most advanced methods to enhance flow boiling heat transfer is to use nanofluids. Nanofluids flow boiling behavior was first described in 2010 by Liu et al. [15] presented both flow boiling enhancement and deterioration of Al_2_O_3_/H_2_O nanofluid. Since then, research on nanofluid flow boiling has been successively published, testing different nanofluids compositions and flow boiling devices with different configurations as reported in the lasted reviews published by M. Saad Kamel (2019) [16] and S. Mukherjee (2022) [17]. The finding of boiling heat transfer enhancement or deterioration has created a controversy that is attracting increasing attention.

Initially, the thermophysical properties of nanofluids were found to be the main factors affecting boiling heat transfer, and further published studies have focused on studying bubble dynamics and flow patterns [18].

The deposition of nanoparticles on the surface has been proposed as a phenomenon that modifies the flow boiling heat transfer coefficient; such deposition can vary the surface roughness; modify the nucleation point density, increasing or decreasing it. However, it has also been postulated that the formation of a layer on the surface increases the thermal resistance, which worsens the boiling heat transfer [19]. Moreover, some results indicate that the liquid adhesion can be enhanced by the capillary force caused by the deposition of nanoparticles, which is the reason why the heat transfer of the nanofluid flow boiling is enhanced [20]. Pinto and Fiorelli [21] concluded that working with nanofluids, the main parameters that directly influence the heat transfer performance are:The types of nanoparticles and their concentration in base fluids.The modification of thermophysical properties, such as thermal conductivity, viscosity, heat capacity, density and surface tension.The modification of wall surface characteristics (i.e., wettability, capillary wicking force surface roughness) during fluid evaporation.The operation condition, especially the mass and heat fluxes for convective flow boiling.

So far, studies on heat transfer in boiling nanofluid flow are scarce because it is a much more complex process than heat transfer in pool boiling. The latest review of the studies has been carried out by Mukherjee (2022) [17], where this fact was evidenced and where it was also shown that although there are some studies on nanofluid boiling heat transfer for micro-channels (<200 µm) and mini-channels (<3 mm), in the case of macro-channels (3–50 mm), the works are very limited.

This same reflection was also observed in Saad et al. [16] reached this same conclusion, noting that there is no experimental database on nanofluids inside conventional tubes with a larger or equal internal diameter (i.e., 19 mm). Likewise, it is noteworthy that the most used configuration in flow boiling has been a horizontal tube of different materials, such as glass, stainless steel or copper [17], and the reports working with vertical tubes being less studied.

Regarding the composition of nanofluids, published studies have evaluated water-based nanofluids [20,22,23,24,25,26] and only a couple of them using hydrochlorofluorocarbon (HCFC) refrigerants, such as R141b and R123 [27,28].

Partially halogenated HCFCs are destined to be banned in the near future due to their environmental impact; an alternative is the use of hydrocarbon-based refrigerants (HC), fluids with low thermal conductivity, such as isobutane (R600a), the predominant refrigerant in domestic refrigerators in Europe, and propane (R-290), which are widely used in heat pumps, air conditioners and commercial refrigeration systems [29]. Reported investigations working with R600 as nanofluid in flow boiling are focused on plain horizontal tube [30,31].

N-pentane has been evaluated as a new base fluid with plausible heat transfer characteristics to enhance the heat transfer properties, binary mixtures based on n-pentane with methanol; acetone has been evaluated by M. M. Sarafraz [32] in a horizontal heat pipe finding that n-pentane was a promising base fluid to promote a higher heat transfer coefficient thanks to promote higher bubbles and its higher vapor pressure. M. Goordazi et al. evaluated the effect of graphene dispersion in an emulsion formulated in a water/n-pentane [33], concluding that the heat transfer coefficient of the emulsion can be increased by the presence of graphene. The higher content of graphene, the higher increment in the heat transfer coefficient.

This article evaluates two main aspects that could cover the gap found in the bibliography: measurement in a vertical tube as an evaporator and a hydrocarbon-based refrigerant, such as n-pentane. The thermophysical properties have been determined and are presented; in addition, the flow boiling heat transfer coefficient has been determined experimentally using a laboratory set-up based on a thermosyphon process generated in a vertical evaporator glass tube of 20 mm in diameter and 1500 mm in length, proposing the heat transfer mechanism governed. The results are correlated with the thermophysical modifications inferred in the refrigerant according to the nanomaterial nature, size, and dispersibility and their interactions with the evaporator surface.

## 2. Materials and Methods

### 2.1. Materials

For this study, commercial nanomaterials (NM), listed in Table 1, of different chemistries, morphologies and sizes have been selected in order to evaluate the effect of these intrinsic properties of the nanomaterial on their dispersibility, as well as on the thermal properties of the final nanofluid.

For the refrigerant, n-pentane (reagent grade, 98% from Sigma-Aldrich) was selected because of its chemical similarity to commercial hydrocarbon refrigerants (such as isobutane R600a and propane R290) and because it is in liquid phase at room temperature, which facilitates its manipulation and characterization for the preparation of the nanofluids.

### 2.2. Nanofluids Sample Preparation

Nanofluids were prepared by a two-step method employing surfactants to increase the dispersibility and stability of NMs in n-pentane. The dispersion protocol was carried out by mechanical agitation for 10 min (Ultraturrax, IKA Staufen Germany, at 1500 rpm), followed by ultrasound for 15 min (VibraCell VCX500-750, Sonics, Newtown, USA- with an ultrasonic probe amplitude of 30% and a frequency of 20 kHz). The sample was kept in an ice bath to avoid temperature increase and evaporation of the sample. Initially, a polymeric nonionic surfactant (Hypermer KD13 from CRODA) was solved in n-pentane, and each nanomaterial was dispersed in the solution in order to obtain 0.5 wt.% of NM. The amount of surfactant in the solution was adjusted in each system to 150 wt.%with respect to the NM. Table 2 summarizes the nanofluids prepared for this work.

### 2.3. Nanofluids Characterization

#### 2.3.1. Effective Nanomaterial Dispersed

The amount of NM effectively dispersed in n-pentane was measured after keeping the samples for 24 h in static storage at room temperature. The sample was separated into two phases (the supernatant and the decanted fraction) and after filtering (0.2 µm filter), washing with n-pentane and drying at 80 °C the decanted phase, the nanomaterial present in the decanted fraction was quantified by weighing. By difference, the amount of nanomaterial effectively dispersed in the supernatant phase was obtained.

#### 2.3.2. Particle Size Distribution

The laser diffraction (LD) technique was used to evaluate the particle size distribution (range 0.02–2000 µm) of nanofluids with a MasterSizer Hydro2000 (Malvern Instruments, Worcestershire, UK). The measurement was performed by adding the nanofluid sample into the dispersion unit filled with n-pentane until a laser light obscuration of 15% was achieved. The dispersion unit kept the mixture stirring at 2000 rpm. Each sample was measured 3 times with 5 s delay between each measurement, and the result was reported as an average of the measurements taken.

#### 2.3.3. Density

The measurements of density (ρ) were performed in an electronic densimeter DX4 equipped with RM40 Refractometer (Mettler Toledo Columbus, OH, USA) following the standard ASTM D4052 at 20 °C of temperature.

#### 2.3.4. Viscosity

The dynamic viscosity of the nanofluids was measured in an MCR101 Rheometer (Anton Paar, Graz, Austria) with concentric cylinder geometry CC27. Measurements were performed under isothermal conditions at 20 °C with a shear rate from 10 s^−1^ to 500 s^−1^. The viscosity values are reported at a 23 s^−1^ shear rate.

#### 2.3.5. Thermal Conductivity

The thermal conductivity was evaluated by hot wire method according to ASTM D5334-08 using KD2-PRO equipment with KS1 sensor (DKGON Meter Group, Inc., Washington, WA, USA). These measurements were carried out at 20 °C in a climatic chamber. The sample was stirred for one minute and held under static conditions for 14 min prior to the experimental measurement. Reported data are the mean of five measurements.

#### 2.3.6. Heat Capacity

The heat capacity (Cp) was measured using Dynamic Scanning Calorimetry (DSC1 model, Mettler Toledo, Columbus, OH, USA) according to ASTM E1269. The method consisted of a dynamic heating ramp from 0 °C to 50 °C at 20 °C per minute under an inert atmosphere of nitrogen at 50 mL/min. The Cp was calculated based on the Sapphire method.

### 2.4. Overall Heat Transfer Coefficient (U)

The determination of the overall heat transfer coefficient (U) has been determined experimentally means a specific laboratory setup that has allowed us to determine the heat transfer coefficient of the refrigerant and the effect of nanoaditivation.

The experimental setup shown in Figure 2 was a two-phase thermosyphon loop consisting of an evaporator, a condenser and a downcomer, which form three different sensorized circuits. All the installed sensors were connected to a PID (proportional, Integral, Derivative) control and to a computer to register and visualize all the parameters. The main elements of the experimental setup and each circuit are detailed in Table 3.

The refrigerant circuit is the core of the experimental setup and is connected to the cold and hot water circuits. The refrigerant circuit consists of a 1500 mm long test tube composed of two concentric transparent glass tubes (evaporator). Two fluids circulate in countercurrent in these two tubes: the refrigerant circulates by force convection in the inner tube in an upward direction, and the hot water circulates in the outer tube in a downward direction. The evaporator’s main characteristics are compiled in Table 4.

The upper part of the inner tube is connected to the condensing chamber (high-strength borosilicate cylinder with an inner diameter of 90 mm, outer diameter of 100 mm and length of 300 mm) where the refrigerant condenses and returns to the evaporator inner through the downcomer.

The cold-water circuit consists of a water-operated thermostatic bath, which pumps cold water to the condenser consisting of a nickel-plated copper tube coil with a surface area of 0.043 m^2^. The hot water circuit operates with water connected to a thermostatic bath (internal resistance of 600 W power) to heat the water and a centrifugal pump to recirculate the hot water from the bath to the cooling circuit (external glass tube).

Both circuits have the necessary sensors to control the variations of the hot water properties. Connected to the refrigerant circuit in the downcomer, there is an extra valve to generate the required vacuum to the system as well as to feed the refrigerant samples into the circuit. Measurements and subsequent calculations are affected by the sensor’s accuracy listed in Table 5.

The overall heat transfer coefficient (U) in a system formed by two concentric tubes can be expressed as the overall heat transfer resistance (1/U), which is a function of the heat transfer resistance of the internal (1/h_i_), external (1/h_ex_) tubes and thermal wall resistance (1/h_wall_), according to Equation (1).
(1)$$\frac{1}{U}≅\frac{1}{h_{i}}+\frac{1}{h_{ex}}+\frac{1}{h_{wall}}$$

Based on glass concentric tubes dimension; diameters and thickeners, (Table 4) and considering water physical properties at 20 °C (Table 6) wall thermal resistance (1/h_wall_), and water thermal resistance (1/h_ex_) can be calculated and consider constants, so the overall heat transfer coefficient (U) can be evaluated by internal heat transfer coefficient (h_i_) which was determined by nanofluids main properties and behavior.

The different experiments were carried out according to the following test protocol.

After charging the refrigerant, dry air was passed through the refrigerant circuit to remove air bubbles. A pump was then used to create a vacuum of 0.6 bar in the refrigerant circuit, and 300 mL of refrigerant sample was added.

As an experimental protocol, the operating conditions of the cold-water circuit were fixed (water flow SQX 1.2 L/min and bath temperature STX 20 °C), as well as the hot water temperature ST1 50 °C. The operating conditions of the hot water flow SQ1 were changed from 0.8 to 2 L/min. In each operating condition, the refrigerant was evaporated in the evaporator glass tube until a stabilized boiling regime was reached, visually checked, and in this steady state, the temperatures of each circuit were recorded. An average pressure value of 0.8 ± 0.5 bar was maintained at the head and 0.70 ± 0.5 bar at the bottom of the evaporator. After the experimental sequence, the refrigerant circuit was drained and cleaned with fresh n-pentane.

## 3. Results

### 3.1. Nanofluids Characterization

The particle size (PZ) distribution of the different nanofluids prepared was measured, and the mean particle value (d0.5) was obtained. Figure 3 shows the obtained distributions.

All the nanofluids show a monomodal distribution, although, in the case of the 0.5 wt.% G20 sample, some bimodality is observed, and in the case of the 0.5 wt.% G200 sample, the size distribution is very wide.

This characterization technique is considered to be exclusively applicable to suspensions containing mainly spherical particles, although, according to previous studies by M K Rabchinskii et al. [34], it was shown that despite the high anisotropy of graphene, the LD method allows determining the lateral mean size of graphene. Experimentally obtained results of d0.5 for graphene-based nanofluid fit with nanomaterial lateral size. The mean particle size of formulated nanofluids in n-pentane is compiled in Table 6.

The critical thermodynamic and thermophysical properties of the refrigerant liquid phase, linked with flow boiling heat transfer characteristic, as reported by A. K. Mozumder et al. [35] are density (ρ), viscosity (µ), thermal conductivity (λ) and specific heat (Cp). Table 7 and Table 8 summarize the effective nanoparticle content dispersed in n-pentane after 24 h and the nanofluids’ main properties in the liquid phase according to real NM content.

It is known that enhancement of the properties of nanofluids depends on the type, size and shape of the nanomaterial, as well as on the particle fraction and particle size distribution [36,37]. The experimental data obtained within the dispersion of the proposed nanomaterials in density was related to the nanomaterial fraction dispersed in the fluid, and the change in this property was negligible. For heat capacity measurements, the effect found was in the same order as experimental error, so these properties were not critical in our case of study.

Li et al. [21] were one of the first researchers to study the transport properties of nanofluids and observed that the viscosity was not only affected by the volume concentration but also by the size of nanoparticles. Subsequently, new experimental studies followed emerged that concluded that the viscosity of nanofluids is dependent on several parameters, such as temperature, volume concentration, aggregation, particle shape, particle size, etc. [24,25]. In this case, experimental data obtained for viscosity at 20 °C show that the increment is not just related to particle fraction and dispersed nanomaterials size and nature; results aligned with other reported results.

For thermal conductivity, one of the proposed and accepted behavior is that thermal conductivity is affected by the Brownian velocity of nanomaterials in the fluid; the higher movement, the higher increment in the conductivity [38], other proposed mechanisms is related to the clustering effect of nanomaterials that increase the hydrodynamic radio promoting a channel for heat transport [39]. The experimental data obtained for the thermal conductivity at 20 °C show an increment in a range of 1–14%, mainly dependent on nanomaterial nature and concentration.

Based on the effective amount of dispersed nanomaterial, it is observed that the Al_2_O_3_-13 nm system was the system with the lowest particle size (180 nm) and the highest content of dispersed nanomaterial, which is why it is considered the most stable nanofluid. This nanofluid also promoted an increase in thermal conductivity of +14% and in viscosity close to 7%. The system formulated with alumina nanowires showed a small amount of effectively dispersed nanomaterial, which is the main reason for discarding this system.

Titanium-based systems were discarded due to their low impact on thermal conductivity (increment in the range of 1.5%), and although the amount of material dispersed after 24 h was 0.3%, the increment in viscosity had the higher effect found (+5%).

Among the graphene-based systems, G200 at 0.5% by weight was selected due to the increment it causes in thermal conductivity (+11%) being more stable in n-pentane than Graphene G20. Table 9 summarizes the selected systems that were tested in the flow boiling experimental setup.

### 3.2. Heat Transfer Coefficient Experimental Determination

There are a large number of correlations available in the literature on the flow boiling of saturated liquids to assess the flow boiling heat transfer coefficient (h_fb_). Chen´s method [40] can be considered the most used correlation for evaporation in vertical tubes, which includes the heat transfer coefficients due to nucleate boiling (h_nb_), characterized by the formation of vapor bubbles on the heated wall, and forced convection boiling mechanism (h_cb_); characterized by the convection through a liquid film on the heated wall and vaporization at the liquid/vapor interface. This correlation uses the Dittus–Boelter correlation for turbulent convection and the Forster–Zuber correlation for nucleate boiling and includes two correction factors, S_f_, known as the boiling suppression factor, and F, the convective enhancement factor. This correlation is simplified in Equation (2) and is based on the thermophysical properties of the fluid and design parameters:h_fb_ = h_nb_ S_f_ + h_cb_ F(2)

From numerous macroscale investigations, it is known that when flow boiling is dominated by the nucleate boiling mechanism, the heat transfer coefficient increases with increasing heat flux (or wall superheat) and saturation pressure and is independent of mass flux and vapor quality. In contrast, in convective-dominated flow boiling, the heat transfer coefficient is independent of heat flux and increases with increasing mass flux and vapor quality [41].

Chen´s equation allows us to determine the heat transfer coefficient in flow boiling for evaporators in vertical tubes, the main limitations of the proposed expression are that (i) surface interactions are not considered and (ii) that it is needed to know physicochemical data of liquid and vapor phases. In the case of a known substance, it might be feasible, but in the case of nanofluids, it is not possible for the moment to characterize the required properties (measure physic properties in the gas phase)

In order to perform the thermal analysis of the nanofluids, the power exchange in the evaporator (q_ev_) at steady state was calculated according to Equation (3), where the power transferred by the hot water (h_w_) will be adsorbed by the refrigerant (r). Additionally, from the flow boiling measurement data, the overall heat transfer coefficients (U) were determined (W/m^2^ °C). This calculation was carried out for different hot water flow rates (Q) with the aim of obtaining a U-flow curve in the evaporator.
(3)$$q_{ev}={\left(ṁCp∆T\right)}_{h_{w}}={\left(ṁCp∆T\right)}_{r}\;≅\left(ṁ∆h_{l\to g}\right)$$

The evaporator is a coaxial tube heat exchanger, and the proposed expression for determining the average overall heat transfer coefficient (and the exchanged power) is based on the Logarithmic Mean Temperature Difference (LMTD).

The overall heat transfer coefficient (U) can be calculated according to Equation (5), where the area of the evaporator (A) is a known data of 0.078 m^2^, and ΔT_lm_ can be conducted according to the nomenclature of Figure 4 for LTMD expression.
(4)$$q_{ev}=UA_{ev}∆T_{lm}$$
(5)$$q_{ev}=UA_{ev}\frac{{(T}_{hw,o}-T_{r,ei})-{(T}_{hw,i}-T_{r,eo})}{{ln((T}_{hw,o}-T_{r,ei})/{(T}_{hw,i}-T_{r,eo}))}$$

Equation (3) can be expressed according to proposed nomenclature in Equation (6), where hot water mass flow rate (m_hw_) can be expressed as hot water flow rate (Q_hw_ in L/min) and water density (ρ in kg/m^3^).
(6)$$q_{ev}={\left(ṁCp∆T\right)}_{hw}={\left(ṁCp∆T\right)}_{r}=Q_{hw}{\rho_{w}}_{}{Cp}_{w}\left(T_{hw,i}-T_{hw,o}\right)$$

Combining Equations (5) and (6) in Equation (7), display the expression applied to experimental determination of U in the evaporator.
(7)$$Q_{hw}{\rho_{w}}_{}{Cp}_{w}\left(T_{hw,i}-T_{hw,o}\right)=UA_{ev}\frac{{(T}_{hw,o}-T_{r,ei})-{(T}_{hw,i}-T_{r,eo})}{{ln((T}_{hw,o}-T_{r,ei})/{(T}_{hw,i}-T_{r,eo}))}$$

Table 10 shows the U values obtained for the evaporator working with n-pentane at proposed water flow rates. External resistance of water (1/h_ex_) was calculated according to Sieder and Tate equation [42] for a laminar flow for each proposed water flow rate and considering wall resistance (Table 4), the internal resistance (1/h_in_) and, thus, heat transfer coefficient of pentane was calculated (Equation (8).
(8)$$\frac{1}{U}=\frac{1}{h_{i}}+\frac{1}{h_{ex}}\frac{D_{in}}{D_{ext}}+\frac{1}{h_{wall}}\frac{D_{in}}{D_{med}}$$

The heat transfer coefficient of n-pentane as reference system is plotted in Figure 5 as a function of Reynolds number.

The heat transfer coefficient of n-pentane tested at different hot water flow rates has shown linearity, being the mean value of pentane was 307 W/m^2^ °C.

Proposed systems, listed in Table 11, based on nanomaterial dispersion in n-pentane, were tested under a defined protocol and compared with n-pentane at different flow rates.

The sample formulated with 0.5 wt.% G20, with Graphene and tested in the flow boiling device (Test T4), did not show a stable boiling pattern in the evaporator tube and the steady state was not reached according to the proposed protocol. Table 12 summarizes experimental data of U in the evaporator at proposed flow rates.

According to Equation (8), the heat transfer coefficient of proposed systems (h_in_) was calculated considering the wall and water resistance. Table 13 summarizes obtained values, and Figure 6 the h_in_-flow rate plot.

The addition of the proposed nanomaterials in n-pentane promoted an increase in the overall heat transfer coefficient in the evaporator (U) due to the increment of the heat transfer coefficient (h_in_) of the refrigerant. It was found that this increment was related to the type of nanomaterial and the concentration of nanomaterial. The alumina nanomaterial tested at 0.01 wt.% and 0.5 wt.% promoted a mean h increase of 12% and 24%, respectively, while graphene nanomaterial tested at 0.01% promoted an average increase of 34%.

Therefore, graphene tested at 0.01 wt.% shows the highest increase in all tested points, with a mean increment of 34%. This experimental behavior is in accordance with the experimental results published by Sarafraz [43], who, after evaluating the boiling heat transfer characteristics in a vertical annulus, a higher heat transfer performance was found for carbon-based nanomaterials. According to experimental data, a graphene-based system evidence an increment in the heat transfer coefficient (h) at higher flow rates. The main hypothesis for this phenomenon could be due to the more intense disturbance caused by the bubbles under a higher heat flux [18].

Although the complexity of the flow boiling heat transfer mechanism, based on experimental data obtained for n-pentane and nanoaditivated systems, it is possible to propose the main components that are governed the flow boiling process. Due to the limitations mentioned in Chen Equation, the convection boiling heat transfer for pentane has been calculated according to Dittus–Boelter [44] expression (h_cb_ = 85 W/m^2^ °C) and, for nucleate boiling heat transfer coefficient Rohsenow expression [45] has been used (h_nb_ = 688 W/m^2^ °C). The experimental mean value obtained for n-pentane (307 W/m^2^ °C) is an intermediate value between the convection boiling heat transfer convection and the nucleate boiling heat transfer and could be expressed according to Equation (9), where ∝ is the fraction of convection and (1 − ∝) the fraction of nucleation.
(9)$$h_{fb}=\propto (h_{cb})+(1-\propto )(h_{nb})$$

According to the experimental value of pentane, the fraction of convention boiling mechanism (∝: 0.63) is dominating. Considering experimental mean values of tested systems where the heat transfer coefficient (h_in_) has been increased due to the presence of nanomaterials, the nucleate boiling mechanism is being promoted. Table 14 shows the estimated values of ∝ for tested systems.

The effect of n-pentane addition with nanomaterials has promoted not only an increment of heat transfer coefficient but also has supported the nucleation mechanism of the process; so it can be concluded that the addition of nanomaterial into n-pentane could modify the dominating boiling heat transfer mechanism [41].

The effect of nanomaterials during flow boiling was also evaluated by means of visual analysis. In fact, the addition of nanomaterials to n-pentane caused three main phenomena that were verified visually. The first one was to generate a process with a higher perturbation, as shown in Figure 7, the other was the generation of more nucleation sites where the bubbles were generated, and the last effect was the deposition of nanomaterials on the glass surface.

This is consistent with previous studies, where it was already shown that in the course of the nucleate boiling process with nanofluids, nanoparticles deposit on the heating surface over time and can modify its properties, including wettability, roughness, and nucleation site density [43]. In particular, the surface roughness after the nanoparticle deposition is influenced by the fraction and intrinsic thermal properties of the nanoparticles and heating surface roughness. The mending effect of nanoparticles in rough surfaces can decrease the density of the nuclear sites decreasing the heat transfer coefficient [46,47], but nanomaterials deposition can promote new and more nucleation points mainly in the polish surface [48], promoting the heat transfer coefficient.

## 4. Discussion

This study proposes the utilization of nanomaterials with different chemistries, including carbon-based materials and metal oxides, such as titanium and alumina, exhibiting various morphologies, such as spherical, wire, and laminar structures, all in nanometric sizes. Among the nanomaterials dispersed in n-pentane, it was found that dispersibility, measured as the remaining quantity of nanomaterials after 24 h, is influenced not only by the nature of the nanomaterials but also by their morphology. With the exception of Al_2_O_3_-13 nm, the other tested nanomaterials showed a low chemical affinity with the host fluid, n-pentane.

The obtained systems exhibited a particle size distribution ranging from nanometric to micrometric sizes, and the experimental results for the thermophysical properties indicate that the dispersion of the proposed nanomaterials has a relatively limited effect. Although the observed variation aligns with previous research findings, the experimental values in this study suggest that the nanomaterials’ dispersion in n-pentane, according to the proposed dispersion protocol using 0.5% of nanomaterials, leads to an approximate 10% increase in thermal conductivity and a penalty in viscosity close to 10%.

To evaluate the overall heat transfer coefficient, an experimental setup was used due to the necessity of conducting experimental tests to evaluate the heat transfer properties in the absence of knowledge about the thermophysical properties of formulated nanofluids in the vapor phase.

The proposed experimental configuration involves a vertical evaporation device consisting of a double-walled glass chamber. This design allows controlled evaporation processes within the chamber. In this configuration, water circulates through the outer tube of the experimental device while the refrigerant flows in an upward direction through the inner tube. The proposed experimental device offers advantages, such as simplicity, transparency for visualization purposes, and the ability to control and manipulate various parameters affecting the evaporation process, including the water flow rate and the choice of the working fluid.

The heat transfer coefficient obtained experimentally for n-pentane and for the proposed experimental conditions have shown a mean value of 307 W/m^2^ °C mainly controlled by convective mechanism. The calculated heat transfer coefficient for n-pentane with nanomaterial dispersion, based on experimental data obtained in the flow boiling device, evidence that:

Nanomaterials dispersion has generated an increment in the calculated heat transfer coefficient, being the increment proportional to the dispersed nanomaterials fraction. Nanomaterials’ nature and their intrinsic properties have shown a higher effect in flow boiling heat transfer coefficient increment than the nanomaterials fraction used, being graphene, the nanomaterial promoting the higher incremental lower quantity. 0.01% of G200 dispersed in n-pentane has promoted a mean increment of 34%, and in the most favorable condition, an increment could be higher than 55% (higher water flow rates).

While n-pentane and alumina-based systems have shown a constant value of heat transfer coefficient independently of water flow rate, the graphene-based system has shown an increment in the heat transfer coefficient with an increment of heat flow; this behavior suggests that the predominant heat transfer mechanism has moved from forced convective heat transfer to nucleate heat transfer. The heat transfer mechanisms have also been proposed through the estimations of fraction coefficients of convection and nucleation.

This behavior has also been visually observed during the experimentation thanks to the glass transparency of the evaporator. N-pentane systems with nanomaterials have promoted higher turbulence in flow boiling and higher nucleation sites.

## 5. Conclusions

In the present work, an experimental investigation was conducted to assess the potential of nanomaterials for enhancing the heat transfer coefficient (h) of pentane as a refrigerant and the following conclusions were made.

Working with nanofluids, it is important to consider and mention not only the particle size of the nanomaterials but also the distribution and mean particle size of generated clusters in the fluid.Measurements of thermophysical properties are not enough to assess the potential of a new refrigerant fluid, and robust tests need to be designed to analyze the boiling process.The design of the test set-up has made it possible to accurately evaluate the effect of pentane and to determine the thermal transfer coefficient of pentane with the addition of nanomaterials of different natures and concentrations.The alumina nanomaterial tested at 0.01 wt.% and 0.5 wt.% promoted a mean h increase of 12% and 24%, respectively, while graphene nanomaterial tested at 0.01 wt.% promoted an average increase of 34%.This improvement cannot be justified by the modification inferred in the thermophysical properties of n-pentane but due to nanoparticles’ behavior under working conditions and their interaction with the working surface.The presence of nanoparticles alters the fluid dynamics and heat transfer mechanisms, leading to enhanced heat transfer performance. These effects go beyond the mere modification of thermophysical properties and involve complex phenomena, such as enhanced nucleation, increased interfacial area, and their interaction with the working surface.

It is crucial to further investigate nanoparticle behaviors and their interactions in order to fully understand and optimize the utilization of nanomaterial dispersions for enhanced heat transfer applications.

## Figures and Tables

**Figure 1 nanomaterials-13-02230-f001:**
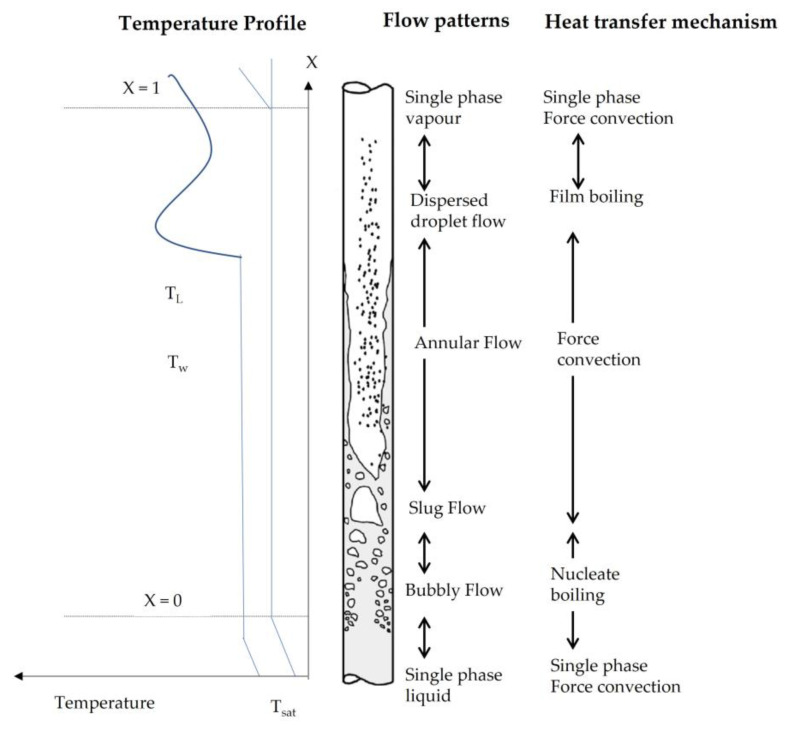
Two-phase flow pattern map of flow boiling and heat transfer mechanisms.

**Figure 2 nanomaterials-13-02230-f002:**
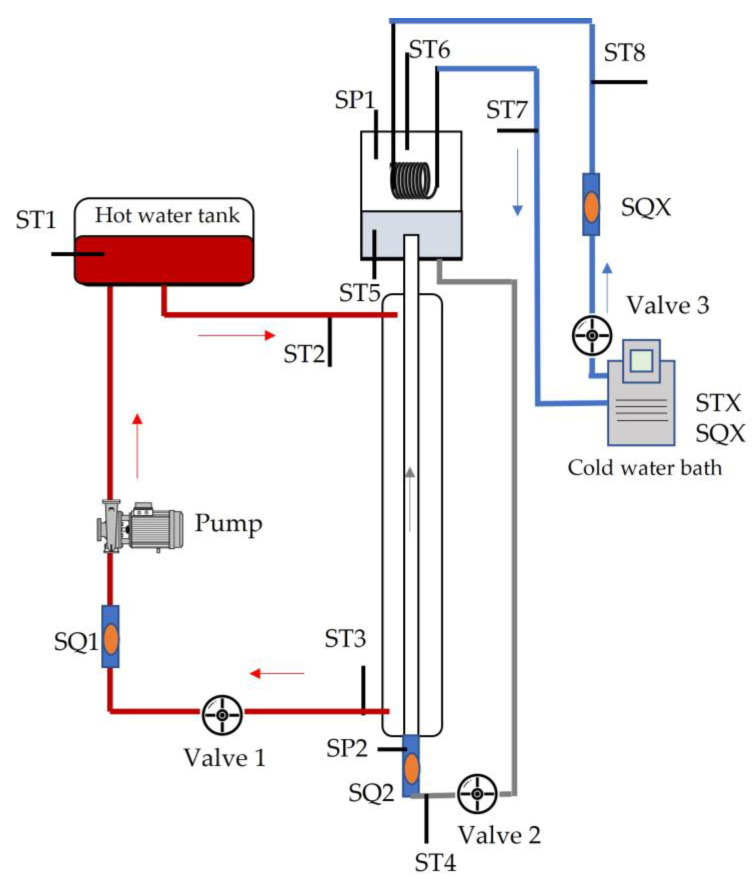
Flow boiling experimental setup.

**Figure 3 nanomaterials-13-02230-f003:**
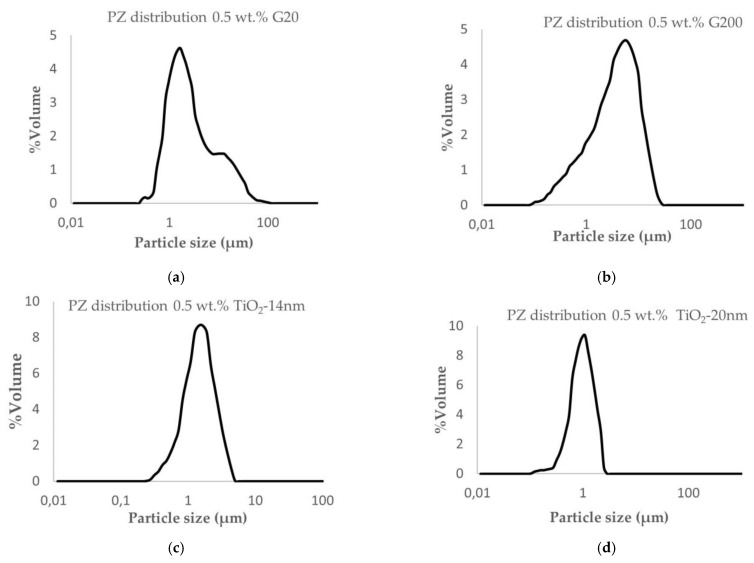
Particle size distribution of prepared nanofluids in n-pentene: (**a**) 0.5 wt.% G20 (**b**) 0.5 wt.% G200 (**c**) 0.5 wt.% TiO_2_-14 nm (**d**) 0.5 wt.% TiO_2_-20 nm (**e**) 0.5 wt.% Al_2_O_3_-13 nm (**f**) 0.5 wt.% Al_2_O_3_-NW.

**Figure 4 nanomaterials-13-02230-f004:**
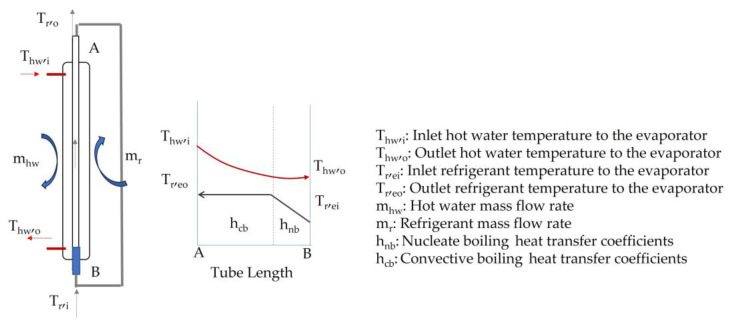
Heat exchange at the evaporator.

**Figure 5 nanomaterials-13-02230-f005:**
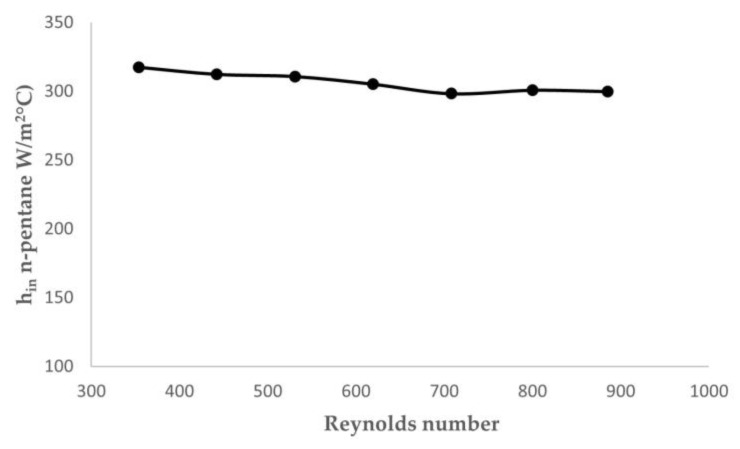
h_in_-Reynolds number plot for n-pentane.

**Figure 6 nanomaterials-13-02230-f006:**
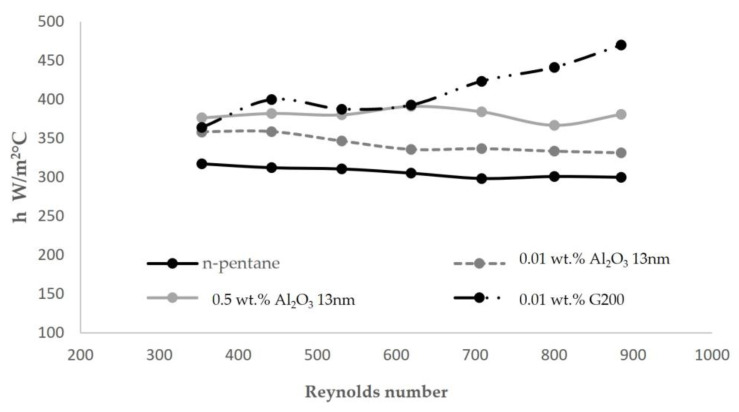
Heat transfer coefficient h_in_ in the evaporator for tested systems in function of Reynolds number.

**Figure 7 nanomaterials-13-02230-f007:**
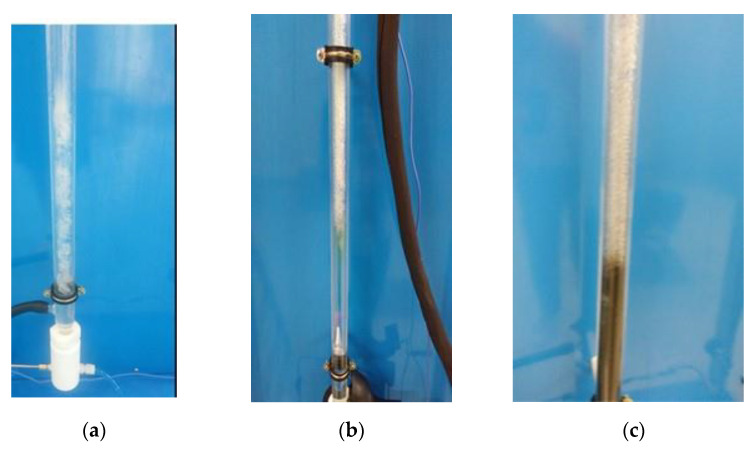
Nucleate boiling process of (**a**) n-pentane, (**b**) alumina dispersion, (**c**) graphene dispersion.

**Table 1 nanomaterials-13-02230-t001:** Selected nanomaterials, main properties, and commercial suppliers.

NM	Morphology	Size	Purity %	BET (m^2^/g)	Density (g/cc)	Supplier
Graphene-G20	Nanopellet	2 μm × 10 nm < 20 layers	99	200		Avanzare
Graphene G-200	Nanopellet	5 μm × 5 nm	98.5		0.2	Avanzare
TiO_2_-14 nm	Spherical	14 nm	99.5	70–110	4.1	Evonik
TiO_2_-20 nm	Spherical	20 nm	99.5	35–65	4.1	Evonik
Al_2_O_3_-13 nm	Spherical	13 nm	99.8	130	3.2	Evonik
Al_2_O_3_-NW	Nanowire	2–6 nm × 200–400 nm				Sigma Aldrich

**Table 2 nanomaterials-13-02230-t002:** Nanofluids prepared in n-pentane for the proposed studio.

Nanofluid Reference	Description
0.5 wt.% G20	0.5 wt.% of Graphene G20 with 150 wt.% of Hypermer KD13
0.5 wt.% G200	0.5 wt.%of Graphene G200 with 150 wt.% of Hypermer KD13
0.5 wt.%TiO_2_-14 nm	0.5 wt.% of TiO_2_-14 nm with 150 wt.% of Hypermer KD13
0.5 wt.% TiO_2_-20 nm	0.5 wt.% of TiO_2_-20 nm with 150 wt.% of Hypermer KD13
0.5 wt.% Al_2_O_3_-13 nm	0.5 wt.% of Al_2_O_3_-13 nm with 150 wt.% of Hypermer KD13
0.5 wt.% Al_2_O_3_-NW	0.5 wt.% of Al_2_O_3_-NW with 150 wt.% of Hypermer KD13

**Table 3 nanomaterials-13-02230-t003:** Elements and codes used for each circuit, (Refrigerant, cold and hot water).

Refrigerant Circuit	
ST4/ST6	Inlet/outlet refrigerant temperature to/from evaporator
SP2	Inlet refrigerant pressure to evaporator
ST5	Outlet refrigerant temperature from condenser
SP1	Refrigerant pressure in the condenser (top)
SQ2	Refrigerant flow
Valve 2	Security valve
**Cold water circuit**	
ST7/ST8	Inlet/outlet temperature to/from the condenser
SQX	Water volume flow rate
STX (set point)	Temperature in the cold-water bath
Valve 3	Valve for cold water flow
**Hot water circuit**	
ST2/ST3	Inlet/outlet water temperature to/from the evaporator
QS1	Water volume flow rate
ST1	Temperature in the hot water bath (setpoint)
Valve 1	Valve for hot water flow

**Table 4 nanomaterials-13-02230-t004:** Evaporator’s main characteristics.

	Outer Tube	Inner Tube
Material	Transparent glass	Transparent glass
Diameter (mm)	32	20
Thickness ε (mm)	2	1.8
Glass thermal conductivity λ (W/mK)	1.1	1.1
Glass thermal resistance $\frac{1}{h_{\mathrm{wall}}}=\frac{\varepsilon }{\lambda }$ (m^2^ K/W)	1.8 × 10^−3^	1.6 × 10^−3^

**Table 5 nanomaterials-13-02230-t005:** The integrated sensor used and accuracy.

Parameter	Sensor Type	Accuracy
Temperature	J type thermocouple	1.5 °C
Pressure	Gauge pressure transduce	0.1%
Volume flow	Rotameter	3%

**Table 6 nanomaterials-13-02230-t006:** Fluids properties at 20 °C, for study case.

Parameter	Water	n-Pentane
Density ρ (kg/m^3^)	1000	630
Viscosity µ (Pa·s)	0.001	0.00023
Heat Capacity Cp (kg/(kJ K))	4.18	2.28
Thermal conductivity λ (W/mK)	0.58	0.11
Prandtl number Pr [$\frac{Cp\times µ}{\lambda }$]	7.15	4.76

**Table 7 nanomaterials-13-02230-t007:** Nanofluids’ physical properties and experimental data with associated error.

System	d(0.5) (µm)	% NP after 24 h	ρ [kg/m^3^] @20 °C	µ [Pa·s] × 10^−4^ @20 °C
n-pentane	-	-	625 ± 0.2	2.31 ± 0.06
0.5 wt.% G20	1.65	0.06	630 ± 0.3	2.46 ± 0.07
0.5 wt.% G200	5.75	0.18	630 ± 0.3	2.54 ± 0.06
0.5 wt.% TiO_2_-14 nm	1.44	0.37	628 ± 0.2	2.45 ± 0.05
0.5 wt.% TiO_2_-20 nm	0.09	0.36	630 ± 0.2	2.42 ± 0.05
0.5 wt.% Al_2_O_3_-13 nm	0.18	0.48	629 ± 0.2	2.48 ± 0.08
0.5 wt.% Al_2_O_3_-nw	0.25	0.09	627 ± 0.3	2.33 ± 0.06

**Table 8 nanomaterials-13-02230-t008:** Nanofluids’ thermal properties and experimental data with associated error.

System	d(0.5) (µm)	% NP after 24 h	λ [W/mK] @20 °C	Cp [kg/(kJ K)] @20 °C
n-pentane	-	-	0.113 ± 0.005	2.15 ± 0.07
0.5 wt.% G20	1.65	0.06	0.122 ± 0.006	2.09 ± 0.05
0.5 wt.% G200	5.75	0.18	0.125 ± 0.008	1.9 ± 0.06
0.5 wt.% TiO_2_-14 nm	1.44	0.37	0.115 ± 0.006	2.3 ± 0.06
0.5 wt.% TiO_2_-20 nm	0.09	0.36	0.114 ± 0.005	2.2 ± 0.06
0.5 wt.% Al_2_O_3_-13 nm	0.18	0.48	0.129 ± 0.005	2.13 ± 0.07
0.5 wt.% Al_2_O_3_-nw	0.25	0.09	0.125 ± 0.006	2.07 ± 0.07

**Table 9 nanomaterials-13-02230-t009:** Selected systems to be tested in the flow boiling setup.

Nanofluid Reference	Description
0.5 wt.% G200	0.5 wt.% of Graphene G200 with 150 wt.% of Hypermer KD13
0.5 wt.% Al_2_O_3_-13 nm	0.5 wt.% of Al_2_O_3_-13 nm with 150 wt.% of Hypermer KD13

**Table 10 nanomaterials-13-02230-t010:** U values for evaporator working with n-pentane at different hot water flows.

Hot Water Flow Rate (L/min)	0.8	1	1.2	1.4	1.6	1.8	2.0
Reynolds number for hot water Re [$\frac{V\times D_{in}\times \rho }{µ}$]	354	442	531	619	708	800	885
U (evaporator) (W/m^2^ °C)	155	157	159	159	159	161	162
1/h_ex_	1.84 × 10^−3^	1.70 × 10^−3^	1.60 × 10^−3^	1.52 × 10^−3^	1.45 × 10^−3^	1.39 × 10^−3^	1.34 × 10^−3^
1/h_in_	3.15 × 10^−3^	3.20 × 10^−3^	3.22 × 10^−3^	3.28 × 10^−3^	3.35 × 10^−3^	3.32 × 10^−3^	3.33 × 10^−3^
h_in_ (n-pentane) (W/m^2^ °C)	317	312	311	305	298	301	300

**Table 11 nanomaterials-13-02230-t011:** Systems tested in flow boiling device.

Test Number	System Reference
T1	0.01 wt.% Al_2_O_3_-13 nm
T2	0.5 wt.% Al_2_O_3_-13 nm
T3	0.01 wt.% G200
T4	0.5 wt.% G200

**Table 12 nanomaterials-13-02230-t012:** Overall heat transfer coefficient (U) in the evaporator (W/m^2^ °C) of proposed systems.

Hot Water Flow (L/min)	n-Pentane	0.01 wt.% Al_2_O_3_-13 nm	0.5 wt.% Al_2_O_3_-13 nm	0.01 wt.% G200
0.8	155	164	168	165
1	157	168	173	176
1.2	159	168	175	177
1.4	159	167	180	180
1.6	159	170	181	189
1.8	161	170	179	195
2.0	162	171	184	202

**Table 13 nanomaterials-13-02230-t013:** Heat transfer coefficients hin (W/m^2^ °C) of proposed systems.

Hot Water Flow Rate (L/min)	0.8	1	1.2	1.4	1.6	1.8	2.0
Reynolds number for hot water Re [$\frac{V\times D_{in}\times \rho }{µ}$]	354	442	531	619	708	800	885
h_in_ n-pentane	317	312	311	305	298	301	300
h_in_ 0.01% wt. Al_2_O_3_-13 nm	358	359	347	336	337	334	333
h_in_ 0.5% wt. Al_2_O_3_-13 nm	376	382	380	391	384	367	381
h_in_ 0.01% wt. G200	364	400	388	393	423	441	470

**Table 14 nanomaterials-13-02230-t014:** Fraction coefficients of flow boiling mechanism.

System	Fraction of Convection ∝	Fraction of Nucleation (1 − ∝)
n-pentane	0.63	0.37
0.01 wt.% Al_2_O_3_-13 nm	0.57	0.43
0.5% wt.% Al_2_O_3_-13 nm	0.51	0.49
0.01 wt.% G200	0.46	0.54

## Data Availability

No applicable.

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
