# Peer review of "Flow Boiling Heat Transfer; Experimental Study of Hydrocarbon Based Nanorefrigerant in a Vertical Tube"

_nanomaterials, 2023, doi:10.3390/nano13152230_

Round 1

Reviewer 1 Report

This study proposes the use of various nanomaterials. There are carbon-based materials and metal oxides suspended in n-pentane. The aim of doing so is to improve the heat transfer coefficient during fluid boiling. The properties of the suspension were evaluated experimentally using a glass vertical evaporator with 20mm inner diameter. The results show that properties show a limited but significant increase in the effect of nanomaterial dispersion to the thermophysical properties of the suspension.

Author Response

Thank you very much for the review, for the feedback on the study that is proposed.

Reviewer 2 Report

This study proposes the use of various nanomaterials; carbon-based materials and metal oxides; in n-pentane as hydrocarbon-based refrigerant, to enhance the flow boiling heat transfer coefficient.

THE WHOLE WORK IS NTERESTING.

POINTS FOR IMPROVEMENT:

1. Please, make a comment regarding pressure drop.

2. Please, use throughout the text dimenionless numbers : Prandtl, Reynolds, etc

3. Please, propose correlations for the heat transfer coefficients

4. A literature review made by the reviewer by using the keywords provided by the authors and GOOGLESCHOLAR revealed that some recent work could be cited.

Author Response

All the comments are in the atached file 

Reviewer 3 Report

Brief summary:

The study deals with the experimental evaluation of the flow boiling heat transfer coefficient of several nanofluids using a vertical evaporation device of glass with an internal diameter of 20mm.

Broad comments:

The Abstract does not describe the real contents of the paper, it should be revised.

The keywords must be revised to be more in correlation with the main goals of the research.

There is no reason to present a detailed theoretical description with an equation in the introduction part, lines 52-71 should be moved out of the introduction and constitute a theoretical description of the problem. The final part of the introduction should be shortened, e.g. like this:

This article evaluates two main aspects that could cover the gap found in the bibliography: the vertical tube measurement and a hydrocarbon-based refrigerant such as n-pentane. The thermophysical properties have been determined and are presented, in addition, the flow boiling heat transfer coefficient has been determined experimentally using a laboratory set-up based on a thermosyphon process generated in a vertical evaporator tube of glass of 20 mm of diameter and 1500 mm length. The results are correlated with the thermophysical modifications inferred in the refrigerant according to the nanomaterial nature, size, and dispersibility.

The theoretical analysis is missing, and the numbering of equations is wrong (eq.2 occurs twice).

Paragraph 3.2 should probably represent the most important (maybe original?) part of the paper, however, it is presented in an unacceptable form. Figure 4 has no informative description, the temperatures in the graph are rotated in the wrong direction. The vertical axis lacks a description. Equations (2)-(8) use letters without any rules – capital letters are changing with the small ones, subscripts with normal letters in brackets – all this without any explanation (cp/Cp; qev/q(ev); q/Q; ∆?lm/∆?lm; d/D). This must be fixed.

The discussion part is confusing and wordy, it does not explain new findings and benefits of the research. The general information is mixed with experimental results and conclusions that are mostly not convincing. A special conclusion part highlighting the most important findings and benefits in a brief form might be helpful.

There are too many formal errors that should be fixed before publication:

·       The units “kg” and “kJ” should be written with a small “k” in all occurrences (Tables 5 and 6).

·       The mark for the unit “liter”, should be written uniformly, i.e. with either small or capital letter, not once this way, once that way.

·       The same is valid for writing the physical characteristics: overall heat transfer resistance (1/U), heat transfer resistance of the internal (1/hi), external (1/hex) tubes, and thermal wall resistance (1/hwall); they should be written using subscripts and either in italics or in  a straight font everywhere, in the text, in the equations, in the tables, or in the figures.

·       Al2O3 should be written with subscripts in all occurrences (l.78, Fig. 3, Fig. 6 e,f, Tab.7, l.299, l.528), as well as TiO2 (Fig. 3 c,d, l.537).

·       The authors should be more resourceful and careful in tables preparation – there is no need to make a special column for standard deviations, they could be written together with the measured value (Tables 5 and 6); if the values are too small the exponent should be included in the heading of the column (Table 5 dynamic viscosity *10-4)

Special comments:

Figure 1              correct “Flow pattenrs” and “Heat transfer mechanims” to “Flow patterns” and “Heat transfer mechanisms”

l.194     “n” is missing in the word “trasfer” in the heading

l.198     “Figure” is missing (“a reference not found”)

The terms used in Table 3 do not correspond with Fig. 2; the following three items are either only in the Table, or in the Figure, or they are different (SP/SP1; - /Valve 1; ST/ - ; QS1/SQ1)

There are two different tables named “Table 3”

Table 5  The dynamic viscosity unit is

Pa·s = (N/m2)·s = ((kg·m/s2)/m2)·s = kg/(m·s)

the unit in the table must be corrected as it is not obvious whether the second is in numerator or in denominator (Pa·s should not cause such problem, of course); moreover, the unit of Heat Capacity is wrong and Prandtl number should be named.

l.234, Table 5 label, 244-5 and eq.8          the sign for degrees of centigrade should be corrected

l.313     correct “Heat trasfer coeficient experiemtal determination” to “Heat transfer coefficient experimental determination”

l.316     the sign for degrees of centigrade should not be written as superscript

Table 11             correct the unit in the label         W/(m °C)

Table 12             there is no need to repeat the unit in the heading in all columns, it should be given only in the first column (Hot water flow rate in L/min)

l.394     correct “rouggness” to “roughness”

l.397     “it can be concluded” (“d” is missing)

l.398     correct “could modified” to “could modify”

There are just a few grammar faults, as a whole English is fine, both in style and grammar. The faults are listed in the Specific comments.

Author Response

All the comments in the attached document

Round 2

Reviewer 2 Report

This work is accepted.

Author Response

Thank you very much for the contributions received and for accepting the manuscript. 

Reviewer 3 Report

Brief summary:

The study deals with the experimental evaluation of the flow boiling heat transfer coefficient of several nanofluids using a vertical evaporation device of glass with an internal diameter of 20 mm.

Broad comments:

The Abstract was improved.

The Keywords are fine already.

The overall structure of the paper seems to be improved sufficiently, most of the errors were fixed.

However, there was probably some misunderstanding of my comments and some errors remain uncorrected.

·       Table 3 should be unified in style: there is no need to write the first third of the first column in bold font and the two remaining thirds in the normal font; there is also no reason to underline Valve 2 itself only and underline Valve 3 and Valve 1 together with the following texts.

·       The unit of Heat Capacity in Table 6 and Table 8 must be corrected, it should be kJ/(kg K); the unit of dynamic viscosity should be preferably Pa·s, to be the same as in Table 7

·       The unit of density in Table 7 must be corrected, it should be g/L (you can compare it with Table 6!!!), multiplication of the last column should be written only in the table header, not in all lines (I mean “µ *10-4 [Pa·s] @20°C”); Al2O3 on the last line should be corrected.

Special comments:

l.333     The power exchanged in the evaporator should be marked just with the subscript “ev” (without brackets, they are useless), therefore it should be more consistent to replace square brackets with round ones and write “…, the power exchanged in the evaporator (qev) at steady state was calculated …”.

Eqs.3,4,5,6         the same type of writing qev should be used as in the previous comment; the same is recommended for the area Aev.

Fig.4      is much better now, but it still can be improved: the temperatures should be written the same way as they are in the following equations (“i”, “o”, “ei”, and “eo”) written as subscripts.

l.336     The sign for degrees of centigrade was not corrected, it is still written as a superscript.

l.375     The label of Fig.5 should be corrected, “in” should be a subscript.

Table 11             correct the unit in the label         W/(m °C)

l.414     the sign “=” should be written in normal font, not as a subscript

l.421     The Greek small letter alpha (U+03B1) should be used in eq.9, not sign “” (U+221D).

l.330     it is not possible for the moment to characterized required properties” it is necessary to delete the redundant “d” at the end of the word “characterize”

l.449     “heat trasfer coeficent” (“n” and “I” are missing)

l.398     correct “could modified” to “could modify”, it was not corrected!!!

Author Response

Thank you very much for the review please i have iincluded in the ne manuscript version 
